# Quantum Bitcoin Mining

**DOI:** 10.3390/e24030323

**Published:** 2022-02-24

**Authors:** Robert Benkoczi, Daya Gaur, Naya Nagy, Marius Nagy, Shahadat Hossain

**Affiliations:** 1Department of Mathematics and Computer Science, University of Lethbridge, Lethbridge, AB T1K 3M4, Canada; robert.benkoczi@uleth.ca (R.B.); gaur@cs.uleth.ca (D.G.); shahadat.hossain@uleth.ca (S.H.); 2College of Computer Science and IT, Imam Abdulrahman Bin Faisal University, Dammam 34212, Saudi Arabia; 3College of Computer Engineering and Science, Prince Mohammad Bin Fahd University, Al Khobar 31952, Saudi Arabia; mnagy@pmu.edu.sa

**Keywords:** Bitcoin, quantum algorithms, quantum security, blockchain, hash functions

## Abstract

This paper studies the effect of quantum computers on Bitcoin mining. The shift in computational paradigm towards quantum computation allows the entire search space of the golden nonce to be queried at once by exploiting quantum superpositions and entanglement. Using Grover’s algorithm, a solution can be extracted in time $O(\sqrt{2^{256}/t})$, where *t* is the target value for the nonce. This is better using a square root over the classical search algorithm that requires $O(2^{256}/t)$ tries. If sufficiently large quantum computers are available for the public, mining activity in the classical sense becomes obsolete, as quantum computers always win. Without considering quantum noise, the size of the quantum computer needs to be ≈$10^{4}$ qubits.

## 1. Introduction

Cryptocurrencies challenge the validity of modern monetary theory, which says that the legal ordinances supported by a government are necessary to gain the acceptance and trust of a currency by the people [1]. Bitcoin does not rely on the support of a government, but on its algorithmic design, together with voluntary human users. Bitcoin’s design has shaped both its success and usage. The security of bitcoin stems from voluntary miners maintaining the integrity of the ledger-blockchain [2]. For extending the blockchain, miners are rewarded with new bitcoins.

Mining in Bitcoin entails the search of bit strings, called nonces. There are two possible nonces in a block. For a description of the block, consisting of the block header and a Merkle tree data structure, see Section 3.1. Nonces appear in the block header and in a leaf of the Merkle tree. Golden nonces, or successful nonces, have to satisfy a condition; namely, the hash value of the block header has to fall within a range. Any change to either of the two nonce values affects the value of the overall hash. When the primary nonce in the block header changes, the hash is easily recomputed. In contrast, changes to the extra nonce, in the leaf of the tree, require the recalculation of parts of the Merkle tree, and thus the recalculation of the hash requires a time complexity dependent on the height of the tree.

Bitcoin mining is a lucrative business. There are large companies that specialize in Bitcoin mining operations. The basic algorithmic approach for mining nonces remains the same: search the nonce values space by trying out possible nonces, either in a predetermined order or in some randomized order. By assumption, there is no bias on golden nonces, and therefore all nonce values are equally likely to be successful. A randomized search is, therefore, the best algorithmic strategy available on a classical computer. To speed up the nonces’ search, significant hardware has been developed. There are four generations of Bitcoin mining hardware: the CPUs, collection of GPUs, FPGAs, and ASICs.

The advantage of using a quantum computer for nonce mining has not been studied exhaustively. Previous studies have assumed that the difficulty of re-hashing remains the same even when the nonce in the Merkle tree is updated [3]. This paper contributes to this direction with a quantum algorithm that uses quantum parallelism to check all possible nonces in superposition at once. The algorithm improves the previous work by utilizing quantum unstructured search to search for the extra nonce, which is more costly. Therefore, the size of the extra nonce and the size of the search space is the miner’s decision. In the quantum algorithm proposed here, the Merkle tree is walked through only once. Additionally, the use of Grover’s search algorithm adds to the search’s complexity, a term on the order of $\sqrt{N}$, where *N* is the size of the search space. This is a theoretical, algorithmic change that speeds up the nonce’s search by a quadratic factor. Present-day quantum computers are far from a physical realization of the algorithm presented here; the restrictions are as follows:The number of qubits is low.The gates and transformations implemented is restricted to the number of inputs and outputs.The circuit size is limited in depth and breadth.

Nevertheless, well-sized quantum computers would consistently outperform classical hardware in search of golden nonces.

The rest of the paper is organized as follows. Section 2 shows previous connections between Bitcoin and quantum cryptography/computation, both as a threat and also some solutions. Section 3 briefly describes the necessary data structures and mining procedures for Bitcoin.

The quantum mining algorithm is described in Section 4. The algorithm has five steps that follow the logical flow of the mining process. Most of these steps rely on both classical and quantum information. The quantum computer’s size and the algorithm’s time cost are analyzed in Section 5. A comparative discussion on the implication of quantum mining in Bitcoin, i.e., quantum supremacy, is discussed in Section 5. Quantum computers are considered to bring about the end of current security systems and solve the problem of security better with quantum security systems. Section 6 concludes the paper.

## 2. Previous Work

Brassard et al. [4] show that hash collisions can be found asymptotically faster with a quantum algorithm than with a classical algorithm. Quantum cryptography attacks, if realized, pose a real threat to today’s security solutions in general.

Post-quantum cryptography refers to the field of cryptographic primitives and systems, which are themselves classical in nature but are resilient to attacks by a quantum computer. SPHINCS [5] is a quantum resilient signature scheme based on hashes. It is also stateless and can be added to existing security schemes. Initially, SPHINCS could provide a low throughput of signatures on a regular CPU but may be improved with the use of classical parallelism [6].

In the domain of finance, quantum approaches dig their way to modelling financial product behavior. Tang et al. [7] designed a quantum circuit to compute the pricing of collateral debt obligations. Collateral debt obligations are financial products based on loans to be sold to investors. The quantum circuit was implemented and analyzed in Qiskit. The evaluation of real life applications of time complexity improvement or accuracy improvement over the classical Monte Carlo simulation remains to be done in a future study.

For Bitcoin itself, Aggarwal et al. [3] discusses quantum attacks on today’s Bitcoin system. Bitcoin has been relatively resistant to proof-of-work for an estimated 10 years. Nevertheless, Bitcoin signatures are more likely to be vulnerable to quantum attacks in an estimated 7 years. Stewart et al. [8] also mention Bitcoin’s signature algorithm, namely the elliptic curve digital signature (ECDSA), as vulnerable to quantum attacks. They also develop a scheme of hardening the Bitcoin transaction signatures, which they claim can be implemented with a soft fork. Another quantum hard solution to overcome the vulnerability of ECDSA is given in [9]. The study of Bitcoin anonymity attacks has been based on classical approaches mostly, due to Zheng et al. [10], Bao et al. [11], Wang et al. [12].

The presence of stale blocks is a known security risk in the Bitcoin blockchain. Selfish mining [13,14] attacks lead to an increase in the number of stale blocks and have been examined by [15] in the context of quantum capable miners.

The unstructured search of nonces in bitcoin is similar to other search problems, such as quantum methods applied to image recognition [16].

## 3. Bitcoin and Quantum Concepts

The algorithms and analyses presented in this paper are from two very different domains: the protocols of Bitcoin are the first domain, and quantum computation is the second. The entire concept of the Bitcoin network and security relies on classical concepts in distributed computing. The Bitcoin network runs on computers and the internet available now. In contrast, quantum computation is a different paradigm. This new paradigm, if commercially available, will unavoidably challenge the status quo of several security technologies. This paper explores a particularly narrow direction, namely, the effect that large quantum computers will have on Bitcoin mining and possibly the entire Bitcoin cryptocurrency.

### 3.1. Bitcoin Data Structures

The ledger of bitcoin transactions is organized in a Blockchain [17]. Each block of the list is a set of transactions organized into a Merkle tree for efficient access and verification [2]. Miners compete to extend the Blockchain for monetary incentives. The computational work done by miners is to find golden nonces in a large search space.

A brief description of the block header and the Merkle tree is necessary; see Figure 1. Each header node has a fixed number of fields. A hash pointer to the Merkle tree structure of transactions, a nonce, and the target value are important. The Merkle tree holds the bitcoin transactions in the leaves, while the internal nodes store the concatenation of the hashes of the two children. The leftmost leaf has a special significance. The node is added by the miner that has created the tree.

There are two nonces for every block in the Blockchain; see Figure 1, the green marked squares. A golden nonce, therefore, is a concatenation of the two nonces.

#### The Nonce

A nonce is simply a bit string with the sole purpose to influence the value of the Hash of the block header. The miner chooses the value of the nonce, and as such, the value does not come from transactions. Recall that, for each newly generated block, there are two nonces: the primary nonce resides in the block header and the second or extra nonce resides in the leftmost leaf of the tree. Bitcoin uses the SHA-256 hash function.

The first nonce that belongs to the block header has a fixed size of 32 bits. The second nonce is of variable size, and the miner can choose the size. The size of the extra nonce is limited by the block size defined by the Bitcoin protocol. The mining effort consists of finding golden nonces, which induce a hash value below the target value specified in the block header. The Bitcoin network determines the target value related to finding a nonce and is updated every 2016 blocks. The difficulty is increased or decreased so that the average time to discover a block (nonce really) is around 10 min.

The hash of the Merkle tree root depends on all the transactions in the block and the extra nonce in the leftmost leaf node. The miners can change the outcome of the hash operation by changing the value of the two nonces. The mining process consists of enumerating nonce values until a suitable pair is found. It is generally believed that there does not exist any better procedure for nonce discovery on classical computers [2].

The difficulty now is that the header nonce values alone do not determine a golden nonce. Therefore, the second nonce has to be populated. The recommended way to search for a suitable pair is to enumerate all values for the header nonce. Then, if no solution is found, a new value is selected for the extra nonce and all the values for the header nonce are tried again. This is a good recommendation as the computation of the Hash for a new value of the extra nonce involves more steps. The second (or the extra) nonce changes the hash value at the leftmost leaf of the Merkle tree, and the intermediate hash pointers have to be recomputed.

Let us examine a bit more carefully which parts of the Blockchain data structure are affected by the change in nonces’ values. The header nonce affects the hash value of the header only. It is used to determine the final hash value of the block header. If only the header nonce is changed, the Hash’s recalculation at the Merkle tree’s root takes constant time. The extra nonce is located in the leftmost leaf node of the bottom of the Merkle tree (see Figure 2). Each intermediate node in the tree stores the Hash of the left child, followed by the right child’s Hash. The hash value of every intermediate node’s right child is permanent as it does not depend on the nonce. Therefore, all the right hashes in the non-leaf nodes are computed once classically. All the nodes in the right subtree of the root contain only nodes with permanent left and right hashes. Consider a node in the left subtree of the root that is not on the tree’s leftmost path. The left and the right child of such a node are also permanent as they do not depend on the nonce. Therefore, the left and the right hashes of all such nodes are also permanent. Thus, only the left hashes along the tree’s leftmost path change whenever the extra nonce changes. If there are *n* transactions (leaves), a change in the extra nonce value requires $\log_{2}n$ hash recalculations to update the hash pointer at the root of the Merkle tree. If *n* is considered a variable parameter, an update of the leaf nonce is asymptotically more costly than an update to the header nonce. A double application of SHA-256 obtains hash values. The hashes’ computations along the leftmost path in the Merkle tree happens sequentially, and the SHA-256 hash function itself is a multistage sequential circuit with a constant depth. We will see later that the quantum algorithm is also sequential for this step.

The part of the Merkle tree that is greyed out in Figure 2 is computed classically and only once when finding nonces. The greyed part of the Merkle tree is called the permanent part. The part that depends on the extra nonce is called the variable part of the tree. In the next section, we will show how the quantum algorithm treats the Merkle tree’s permanent part to be classical in nature. More interestingly, the variable part of the Merkle tree is treated as quantum in nature.

### 3.2. Notation

The circuit that we develop in the next section has several stages. Each stage passes some of its output qubits to the next stage as input qubits. We describe the circuit in each stage using unitaries as much as possible. At times we will use density matrices obtained using partial trace to describe the appropriate quantum subsystem. Please see [18] for notation and definitions. We will also use quantum circuits for Grover’s search [19] and the SHA-256 [20] hash function. For the implementation of Grover’s search, see [21].

## 4. Quantum Algorithm for Bitcoin Mining

The only possibility to generate new bitcoins is by participating in the maintenance of the Blockchain, the history of all transactions, which keeps the currency alive. Mining involves searching for the two nonces (header and the extra nonce), such that the Hash of the block header is below the target specified. (Figure 3)

We suppose that we have a quantum computer with sufficient qubits to perform all the operations. We will separate the classical part of the circuit needed to compute the permanent part of the Merkle tree and for the application of SHA-256. This separation yields a hybrid solution; the quantum computer needs fewer qubits and works together with a classical computer. There are five stages, and the circuit description for each stage is next.


**Step 1: Compute the permanent part of the Merkle tree classically**


The algorithm computes all the permanent values at the start and only once. As shown in Section 3.1, whenever the nonces change values, only a few nodes in the Merkle tree change. Consider the tree shown in Figure 2. The tree has *n* leaves, with transactions in leaves ordered from left to right $T_{0}$, $T_{1}$, …, $T_{n}$. The first leaf $T_{0}$ contains the nonce, but the other transactions are permanent. The fields affected by the change of the nonces are marked with a red circle.

Given the n−1 transactions in the tree, we compute classically, using post-order traversal, all the hashes at intermediate nodes that they determine. The result of this step is an array of size at most *n* that contains all the right hashes in the Merkle tree, including the ones on the leftmost path. The left-hashes on the Merkle tree’s leftmost path are computed using a quantum algorithm that will be described later.

The hash function used by Bitcoin (HASH) is the double application of SHA-256. Each non-leaf node stores the concatenated Hash of its children. All the hash values computed as above are used in Step 3. We know the left and the right hash for every non-leaf node that does not lie on the leftmost path. For nodes that are on the leftmost path, we know only the value of the right hash. The hash values are computed using a post-order traversal, and the time it takes is proportional to the size of the Merkle tree (O(n)).


**Step 2: Prepare the leaf nonce in a quantum superposition and compute the first Hash.**


The next step is to compute the hash values at the parent of the leaf $T_{0}$, containing the nonce. This hash value is computed using a sequence of quantum circuits, as shown next.

In the first stage, we prepare the input to the quantum circuit. We treat each state as equally likely and consider a superposition of all the states. The number of bits in extra nonce is $n_{2}$. A quantum register of size $n_{2}$, that is a uniform superposition of all possible binary values, is prepared. This is accomplished with an application of $n_{2}$ Hadamard gates applied to a register with value zero, as shown in Figure 4, on the first horizontal line.

In the second stage, we consider the data in leaf $T_{1}$. The nonce register is concatenated to a possibly classical register ($T_{1}$) that holds the miner’s transaction data. The miner’s data refers to the miner’s Bitcoin identity, the address, and the value in bitcoins rewarded for the block creation. This data is classical yet inputs into a quantum circuit. After concatenation of two registers, the set of all possible states is shown by tensor product, denoted by ⊗. The circuit in Figure 4 has 256 additional input qubits. These inputs have the value |0〉 on the left (input) side of the circuit and will carry the value of the HASH on the right side of the circuit, the output.

The input remains unchanged in the output. HASH itself has a quantum implementation [22]; however, in our later analysis, we consider a classical circuit for HASH.

The third stage is the application of the function HASH. HASH allows input of an arbitrary size but always outputs 256 bits. This circuit has constant depth [20].

The input after the application of the Hadamard gates is
$$S_{0}=|T_{1}\rangle ⊗H^{⊗n_{2}}|0_{1}0_{2}\ldots 0_{n_{2}}\rangle ⊗|0_{1}0_{2}\ldots 0_{256}\rangle =|T_{1}\rangle ⊗\frac{1}{\sqrt{2^{n_{2}}}}\sum_{i=0}^{2^{n}-1}\left|i\rangle ⊗\right|0_{1}0_{2}\ldots 0_{256}\rangle .$$

The output ($S_{1}$) of this state is an entanglement of the extra nonces, the miner’s transaction in $T_{1}$, and the hashes of all nonces. If we denote by $hash(i,T_{1})$ the hash value at the parent of $T_{1}$, then the state of the quantum system after the application of HASH is
$$S_{1}=|T_{1}\rangle ⊗\frac{1}{\sqrt{2^{n_{2}}}}\sum_{i=0}^{2^{n}-1}|i\rangle ⊗|hash\left(i,T_{1}\right)\rangle .$$

The next step computes the remaining hashes on the leftmost path in the Merkle tree.


**Step 3: Compute all the hashes on the leftmost leg of the Merkle tree.**


The hash values are computed sequentially in a bottom-up fashion. Instead of computing the hash values at each node for every nonce input sequentially, we again rely on quantum parallelism. With a single application of a quantum circuit, we compute in parallel all the hash values for all possible nonce inputs at a given node. The number of such stages is proportional to the number of nodes in the leftmost path.

The initial input is part of the entangled superposition $S_{1}$, which is concatenated with the right-hash of the internal node at level $\log_{2}n-1$. This right-hash is already computed in Step 1. HASH is applied to this input. A 256 qubit register (initialized to 0) is used for the output. This circuit is again guaranteed to exist, as discussed in Section 3.2.

Figure 5 shows the operation in the first stage on the left. From here on, the hash output of the previous level is the input to the HASH circuit of the next level (as the Hash of the left child). Each sequential circuit has the same three inputs: the hash of the previous level, the hash of the right child, and a register of 256 zeroes to hold the result. After $\log_{2}n$ levels, the hash of the Merkle tree root is the output. The output is in a superposition state, and for every state (nonce value) *s*, we have HASH(*s*). Note that at each level, some qubits are discarded.

The useful part of the output is the initial superposition of all nonces and the superposition of all HASH(s) at the root of the Merkle tree, shown in green in Figure 5. We denote this useful output by $S_{2}$. After this state, the complete state of the system, $\rho_{2}$, is an entanglement of all useful and non-useful qubits. $S_{2}$ can be described only by a density matrix, obtained from $\rho_{2}$ after tracing out the unused qubits. Using the notation in [18] for a subsystem, we have
$$S_{2}=tr_{\mathrm{unused}}\left(\rho_{2}\right)$$

The next step computes the hash of the 4-tuple, given the nonce in the block header. The subsequent step is the use of Grover’s algorithm. The final step is the measurement, which identifies the two nonce values with sufficiently high probability (Figure 6).


**Step 4: Computation of the final hash, given the nonces in the block header.**


This step deals only with information from the block header. It works with two quantum registers in superposition. The first quantum register is the output of the previous step, $S_{2}$. The second quantum register is the superposition of all header nonces. The header nonce is of fixed size, 32 bits. Therefore, it can be represented as a 32 qubit register holding all primary nonce values in superposition. The superposition is generated in the usual way with Hadamard gates.

The hash computed in Step 3 is fed into a HASH (SHA-256x2) circuit and the rest of the header fields. This step is a milestone of the quantum computation as it now has all the possible hash values in a single 256 qubit quantum register. This step’s useful output is an entangled superposition of three quantum registers: all possible hashes, together with the primary nonces of the header, and the extra nonces from the leaf of the Merkle tree. This state can be described as a mixed state of the entire system. If the system at this stage is denoted $\rho_{3}$, then the useful output $S_{3}$ is the mixed state of the three quantum registers after the unused qubits have been traced out.
$$S_{3}=tr_{\mathrm{unused}}\left(\rho_{3}\right)$$


**Step 5: Unstructured Search—Grover’s algorithm.**


The input $S_{3}$ to this step is the hashes’ entire search space in the block header. In $S_{3}$, every pair of the primary nonce with the secondary nonce is entangled with the block’s hash.

In particular, the size of the search space is given by the three registers: the header-nonce is of size 32, the size of the extra nonce, $n_{2}$, and the block-hash is of size 256. The hash’s size does not contribute to the search space, as it is dependent on the nonces. The two types of nonces are independent and thus define the search space’s size, which is $2^{32+n_{2}}$. The state given by these three registers is mixed but describes the system as seen by the Grover circuit in this step, $S_{3}=S_{3}\left(\mathrm{header}\_\mathrm{nonce},\mathrm{leaf}\_\mathrm{nonce},\mathrm{bolck}\_\mathrm{hash}\right)$

Most of the components in the state $S_{3}$ do not contain the golden nonce. However, there is at least one golden nonce among them, extracted by Grover’s algorithm, with a sufficiently large probability. Grover’s algorithm enhances the probability of the solution. Otherwise, from a uniform superposition of the entire search space, the solutions’ coefficients become arbitrarily larger than those of the non-solutions. We define a conditional NOT inversion operator $U_{\omega }$ that inverts only components that represent a solution. This operator acts on the block hash |h〉, together with an ancilla qubit. The ancilla qubit is a boolean value that will hold a superposition, one for a solution, and zero otherwise, see Figure 7. The transformation $U_{S}$, the reflection operator, acts as a definite increase to the solution’s amplitude and reduces the amplitude of the non-solution components. These two transformations are applied iteratively for $\sqrt{2^{256}/t}$ times, where t≥1 is the number of solutions combined and is the probability of all solutions to approach 1.

The state after this step, the resulting state of the system, is $S_{4}$. It is similar to $S_{3}$ in structure, and the amplitudes of the solutions are larger in magnitude (arbitrarily close to 1).


**Step 6: Measurement and interpretation.**


The final step is to measure the qubit registers to determine the header nonce’s value and the extra nonce. The measurement collapses the state, and we get classical information back. As Grover’s algorithm has amplified the states’ amplitudes that correspond to a solution, a measurement will collapse the system to a solution with high probability. As in all quantum algorithms, only one solution can be extracted. The others will be forever forgotten after the collapse. Several repetitions of the quantum algorithm can be performed to boost the classical probability of success. The state after this step will be
$$S_{5}=\mathrm{extra}-\mathrm{nonce}+\mathrm{primary}-\mathrm{nonce}+\mathrm{hash}-\mathrm{of}-\mathrm{the}-\mathrm{block}+1.$$
where + denotes concatenation of strings. Note that the last 1 denotes the success of the findings. In the unlikely case where the last bit is 0, it means that the nonces in the result are actually not solutions.

The resulting nonce values can now be tested classically, and they should be good. The header nonce and the leaf nonce are inserted in the new block, and the block is ready for broadcasting to the network peers. This ends the general description of the quantum algorithm for finding the header and the leaf nonce. Let us apply the steps on a small example.

### A Small Example

To see how the algorithm is applied, let us consider a small, degenerate example. We will consider the smallest possible Merkle tree and the smallest nonces; see Figure 8. Let the Merkle tree have two leaves only and let both nonces be of size 1; one qubit. In this case, the nonce search space is of size $2^{2}=4$, namely nonce=leaf_nonce+header_nonce∈{00,01,10,11}. For definiteness, let us consider that the golden nonce is 01. This value is chosen as a proof of concept and not resulting from any calculation.

The target value in the header is the size of a SHA-256 output, namely 256 bits.

The following transformations show a simplified application of the algorithm’s steps.


**Step 1: Compute the permanent part of the Merkle tree classically.**


The tree has only one transaction *t*. This transaction is hashed twice with SHA-256. For simplicity of notation, we will denote ${\mathrm{SHA}}^{256}({\mathrm{SHA}}^{256}\left(\ldots \right)=\mathrm{SHA}\left(\ldots \right)$. The computation in this step is
rightChild=SHA(t)


**Step 2: Prepare the leaf nonce in a quantum superposition and compute the first hash.**


The extra nonce is now prepared in a superposition of all possible values. In our case, with nonces of size one, this is a simple superposition of 0 and 1.
$$leaf\_nonce=\frac{1}{\sqrt{2}}(|0\rangle +\left|1\rangle \right)$$

For simplicity, ignore the miner’s transaction. Then, the state $S_{0}$ can be written as
$$S_{0}=\frac{1}{\sqrt{2}}(|0\rangle +|1\rangle )⊗|0_{1}0_{2}\ldots 0_{256}\rangle .$$

In this step, the hash of this leaf is computed:$$\mathrm{leftChild}=S_{1}=U_{SHA}S_{0}=\frac{1}{\sqrt{2}}|0\rangle \;|\mathrm{SHA}\left(0\right)\rangle +\frac{1}{\sqrt{2}}|1\rangle \;|\mathrm{SHA}\left(1\right)\rangle$$
where $U_{SHA}$ is the quantum transformation that implements SHA-256 twice.


**Step 3: Compute all the hashes on the leftmost leg of the Merkle tree.**


For our example, this step is degenerate, as the height of our tiny tree is one. The new state is the simple concatenation of the output of the previous two steps.
$$S_{2}=S_{1}\;|\mathrm{SHA}\left(t\right)\rangle =\frac{1}{\sqrt{2}}|0\rangle \;|\mathrm{SHA}\left(0\right)\rangle \;|\mathrm{SHA}\left(t\right)\rangle +\frac{1}{\sqrt{2}}|1\rangle \;|\mathrm{SHA}\left(1\right)\rangle \;|\mathrm{SHA}\left(t\right)\rangle$$


**Step 4: Computation of the final hash, given the nonce in the block header.**


This step computes the final hash. The final hash is applied on the header. The header consists of

The quantum hash of the Merkle tree, $S_{2}$The primary nonce, in superposition of all possible values. This is similar to the initial superposition of the leaf nonce.
$$header\_nonce=\frac{1}{\sqrt{2}}(|0\rangle +\left|1\rangle \right)$$Some additional classical information: the target value and some identification information for the block. For simplicity, we denote this information with header_info, and though classical, it needs to be fed into the circuit as quantum values, |header_info〉.

The input to step 4 also contains the qubits for the value of the hash $|0_{1}0_{2}...0_{256}\rangle$. The result of this step is obtained as follows:$$S_{3}=U_{SHA}\left(S_{2}\;\frac{|0\rangle +|1\rangle }{\sqrt{2}}\;|header\_info\rangle \;|0_{1}0_{2}...0_{256}\rangle \right)$$
$$=U_{SHA}\left(\frac{1}{2}\left(|0\rangle \;|\mathrm{SHA}(0)\rangle \;|\mathrm{SHA}(t)\rangle +|1\rangle \;|\mathrm{SHA}(1)\rangle \;|\mathrm{SHA}(t)\rangle \right)\left(|0\rangle +|1\rangle \right)\;|header\_info\rangle \;|0_{1}0_{2}...0_{256}\rangle \right)$$

To get a term by term formula, we rearrange the terms of the superposition.



$S_{3}=U_{SHA}\left(\frac{1}{2}\left(|00\rangle \;|\mathrm{SHA}(0)\rangle +|01\rangle \;|\mathrm{SHA}(0)\rangle +|10\rangle \;|\mathrm{SHA}(1)\rangle +|11\rangle \;|\mathrm{SHA}(1)\rangle \right)\left|\mathrm{SHA}\left(t\right)\rangle \;|header\_info\rangle \;\right|0_{1}0_{2}...0_{256}\rangle \right)$



After applying $U_{SHA}$, we get:$$\begin{array}{c} S_{3}=\frac{1}{2}(|00\rangle \;|\mathrm{SHA}(0)\rangle \;|\mathrm{SHA}(00)\rangle +|01\rangle \;|\mathrm{SHA}(0)\rangle \;|\mathrm{SHA}(01)\rangle +|10\rangle \;|\mathrm{SHA}(1)\rangle \;|\mathrm{SHA}(10)\rangle \\ +|11\rangle \;|\mathrm{SHA}(1)\rangle \;|\mathrm{SHA}(11)\rangle )\;|\mathrm{SHA}(t)\rangle \;|header\_info\rangle \end{array}$$

In the above formula, |SHA(00)〉 represents the application of $U_{SHA}$ on the first term:$$|\mathrm{SHA}\left(00\right)\rangle =U_{SHA}\left(\left|00\rangle \;|\mathrm{SHA}\left(0\right)\rangle \;|\mathrm{SHA}\left(t\right)\rangle \;|header\_info\rangle \right|0_{1}0_{2}...0_{256}\rangle \right)$$

|SHA(01)〉, |SHA(10)〉, |SHA(11)〉 have similar meanings.


**Step 5: Unstructured Search—Grover’s algorithm.**


Step 5 applies a standard Grover transformation on $S_{3}$. In order to do this, we need to concatenate $S_{3}$ with an ancilla qubit $q_{a}=\frac{1}{\sqrt{2}}(|0\rangle +\left|1\rangle \right)$ and then apply Grover transformation, to be denoted here as *G*. The result is an amplification of the coefficient of golden nonces.
$$\begin{array}{c} S_{4}=G\left(S3q_{a}\right)=\alpha |00\rangle \;|\mathrm{SHA}(0)\rangle \;|\mathrm{SHA}(00)\rangle \;|\mathrm{SHA}(t)\rangle \;|header\_info\rangle |0\rangle \\ +\beta |01\rangle \;|\mathrm{SHA}(0)\rangle \;|\mathrm{SHA}(01)\rangle \;|\mathrm{SHA}(t)\rangle \;|header\_info\rangle |1\rangle \\ +\gamma |10\rangle \;|\mathrm{SHA}(1)\rangle \;|\mathrm{SHA}(10)\rangle \;|\mathrm{SHA}(t)\rangle \;|header\_info\rangle |0\rangle \\ +\delta |11\rangle \;|\mathrm{SHA}(1)\rangle \;|\mathrm{SHA}(11)\rangle \;|\mathrm{SHA}(t)\rangle \;|header\_info\rangle |0\rangle \end{array}$$

As the coefficients are probabilities, they obey the condition ${\left|\alpha \right|}^{2}+{\left|\beta \right|}^{2}+{\left|\gamma \right|}^{2}+{\left|\delta \right|}^{2}=1$. Notice that, for the second term, the ancilla qubit is |1〉, showing that 01 is the golden nonce. According to Grover, the respective probability coefficient is large, ${\left|\beta \right|}^{2}{>>|\alpha |}^{2}{,|\gamma |}^{2},{\left|\delta \right|}^{2}$.


**Step 6: Measurement and interpretation.**


In the final step, the state is simply measured and will collapse to one of the terms of the superposition. Because the second term has a large probability coefficient, the state of the system will most likely collapse to
$$S_{5}=|01\rangle \;|\mathrm{SHA}(0)\rangle \;|\mathrm{SHA}(01)\rangle \;|\mathrm{SHA}(t)\rangle \;|header\_info\rangle |1\rangle$$

The golden nonce is visible in the first two qubits of the state. Additionally, the ancilla qubit works as a check for the correctness of the nonce.

The question remains: what is the gain in speed or time complexity, realized by this quantum algorithm over the classical algorithm?

## 5. Size and Cost of the Quantum Solution

The algorithm described in the previous section is a hybrid one. It uses both classical and quantum information. Information such as the customers’ transactions and hash values at the nodes other than the Merkle tree’s leftmost path is all classical information. Nevertheless, the right-hash values (which are classical) along the leftmost path in the Merkle tree do participate in the computation of the hash’s quantum superposition at the root of the tree. Thus, this information has to be fed into a quantum circuit.

Quantum circuits in quantum computers so far explicitly expect qubits as inputs and provide qubits on the output. The output qubits are measured to provide an answer as a binary string. Our analysis will suppose that we have sufficient qubits in the quantum computer to hold the complete input (including the classical information) to quantum circuits.

The primary purpose of developing the quantum algorithm is to outperform classical miners. The search space of the nonces is $2^{32}\times 2^{n_{2}}$. The mining process on a classical computer is memoryless, so the exhaustive randomized search is performed by the miners [17]. The mining difficulty is a measure of the time that miners need to find a nonce, which, on average, is about 10 min for the Bitcoin Blockchain.

The situation with the quantum algorithm is very different because of quantum parallelism. For the classical part of the algorithm, we use the worst-case running time as a measure. For the quantum part, we use the depth of the quantum circuit as a measure of complexity.

1.*Step 1.* This step is a classical computation on the Merkle tree. Each node is traversed a constant number of times. The nodes on the leftmost leg of the tree are processed in the next steps. The number of leaves is *n*. Thus, the computation of this step is a classical Θ(n).2.*Step 2.* This step works on one single leaf node (transaction node) with a quantum superposition. It takes constant time, O(1).3.*Step 3.* This step computes the superposition of the hash values along the leftmost path of the tree. The tree has logn levels, and each level takes constant time. The overall execution time for this step is Θ(logn).4.*Step 4.* This step computes the superposition of the hash values in the header. The overall execution time for this step is constant, O(1).5.*Step 5.* This step is an application of Grover’s algorithm on the superposition of all hashes of the block. As the hash is of length 256 and there are *t* solutions, this steps takes $\Theta (\sqrt{2^{256}/t})$.6.*Step 6.* This step performs a constant time measurement. Additionally, but not necessarily, the step may check the Hash with a classical algorithm. The execution time is a constant, O(1).

The most costly steps in the above algorithm are step 3 and step 5. We can say that the overall complexity of the quantum algorithm is $O(\log n+\sqrt{2^{256}/t})$, where *n* is the number of transactions, 256 is the size of the output of HASH, and *t* is the number of golden nonces. We may reasonably consider that $n<<\sqrt{2^{256}/t}$, the number of transactions, is significantly smaller than the search space of the nonces. Thus, the time complexity becomes $O(\sqrt{2^{256}/t})$. This is a quadratic improvement over the classical complexity discussed next.

A crude estimate of the search space is given by the total size of two nonces, $n=n_{1}+n_{2}=32+n_{2}$. If we assume that the header nonce alone is sufficient to provide a right hash, then, as SHA-256 returns a 256-bit output under the standard cryptographic assumptions, the expected number of hashes to be tried is $2^{256}/t$, where *t* is the target or the number of solutions. The difficulty of finding a header nonce classically can be computed from the target value described in Section 3. If the extra nonce field is also used to determine the nonce values, then the overhead associated with the computation of hash values at the intermediate nodes in the Merkle tree has to be taken into account.

To the best of our knowledge, no analysis of difficulty (either classically or quantumly) addresses this. Ours is the first study investigating the difficulty of computing the intermediate hashes in the Merkle tree in a hybrid algorithm. Compared to Grover’s algorithm’s straightforward application (discussed above), the algorithm in this paper exhibits a quadratic improvement.

### Quantum Supremacy

For the moment, the results above have to be categorized as theoretically posing a challenge to Bitcoin but not really feasible in practice. The reason for this is the modest size of quantum computers available right now. Currently, quantum computers freely available to researchers have 5–15–50 qubits. Up to 100–150 qubits are expected to be available soon, meaning in 2 years approximately [23].

The size of the quantum computer necessary for implementing our algorithm is theoretically two orders of magnitude larger. The size of the combined nonce is $n_{nonce}=32+n_{2}$. The most space consuming registers are the registers that hold the 256-bit hash values. As quantum circuits are information preserving, all intermediate quantum hash values, that is, the left leg’s values, are preserved throughout the quantum computation. The total size of all these hashes is 256×(logn+2). Additionally, there is the classical information that needs to be fed into the quantum circuit as qubits. These are:1.The hash values for the right children of the left leg, the result of step 1, which has a size of 256×logn.2.Some additional classical information from the miner’s transaction and the header of the tree. These are of constant size; let us denote this constant with *k*.

The total number of qubits necessary is therefore:$$\mathrm{Total}\;\mathrm{number}\;\mathrm{of}\;\mathrm{qubits}=32+n_{2}+256\times \left(2\log n+2\right)+k$$

We upper bound all parameters in the formula above. The value of any hash is 256 bits long. Therefore, the golden nonce need not be any longer than $256\geq 32+n_{2}$ qubits. For a maximum of $n_{2}=256$−32=224, the expected probability to find one unique solution is 1. We upper bound *n* by the maximum number of transactions a block can hold. The allowed size of the block defines this number. The block’s size has a rather dramatic history, as the initial size of 1 MB proved to be insufficient in the long run. As the size could not be altered except by a hard fork, since 2018, Bitcoin has had two rival chains: one with a block size of up to 32 MB, the other size up to 2 GB. Overall, we may consider a block to hold anywhere between 1500 to 3000 transactions. Thus, n≤3000 and logn=11.

Parameter *k* is the size of the header plus the size of the transaction from the miner. The size of the header, which is an element of the ledger, is insignificant. The size of the miner’s transaction is also generally small. For now, we consider *k* as insignificant in comparison to the other values. The problem arises when, to overturn the quantum computer’s power, the miner’s transaction is made large artificially. In this case, there may be a race between the size of the quantum computer and the transaction size. Nevertheless, as Bitcoin is working in reality, we may consider *k* as small to the point of insignificant.

Thus, the overall size of the quantum memory has to be approximately
$$\mathrm{Total}\;\mathrm{number}\;\mathrm{of}\;\mathrm{qubits}\approx 256\left(22+3\right)+k=8448+k\approx 10^{4}.$$

The number of bits needed is large for a quantum computer but small for a regular computer. Under the assumption that quantum computers may exhibit a growth comparable with classical computers, this number may not be so far in sight.

An open problem is the depth of the circuit necessary in the algorithm. Here, the evaluation is unclear, as the depth of the $U_{\omega }$ transformation and the SHA-256 circuit is not easy to evaluate.

The above analysis is based on ideal qubits within deep depth circuits. In the presence of noisy qubits, the evaluation of the number of qubits necessary must necessarily increase. Zhou et al. [24] give a theoretical evaluation for six types of noises: bit flipping noise, phase flipping noise, bit-phase flipping noise, depolarizing noise, and amplitude damping noise. In all these analyses, the probability of a noise event to occur is a parameter of the calculation. This probability depends on the implementation of qubits, is also in rapid technological progress, and is therefore not easy to evaluate within our algorithm. In addition to the influence of noise, there is also a small probability of failure in the Grover algorithm itself, which can be solved by replacing the phase inversions with some special phase rotations given in Long [25]. The detailed performance of the quantum search algorithm has been extensively studied in Toyama et al. [26].

## 6. Conclusions

We have shown in this paper that the arrival of large quantum computers poses a significant challenge to the Bitcoin Blockchain. Under the assumption that quantum circuits and integrated circuits are comparable in speed, miners with quantum computers shall readily outperform all classical computation approaches.

We have designed a quantum algorithm that computes the nonces necessary for mining a Bitcoin block. The main contribution of our work is algorithmic. To the best of our knowledge, this is the first attempt at the description of a quantum circuit that details the steps needed on Bitcoin data structures (specifically Merkle tree) to find the nonces. We take into consideration both the nonces, the header nonce, and the extra nonce. We also give a logical-level circuit description of the algorithm.

Nonce finding algorithms, either classical or quantum, deal with a large search space that is upper bounded by a constant $2^{256}$. The target value *t*, given in the block’s header, reduces the search space to $2^{256}/t$. A classical algorithm has to search $O(2^{256}/t)$ steps, while our quantum algorithm takes $O(\sqrt{2^{256}/t})$ Grover iterations. This square root improvement is significant because a classical computer cannot compete with the quantum computer, which always wins in mining the next block.

It is an open problem as to whether the limit of 256 for the hash’s size will continue to be large enough for a quantum miners’ race. Otherwise, a new hashing function has to be employed in the Bitcoin structure. This would be a major change in the system, probably a hard fork.

## Figures and Tables

**Figure 1 entropy-24-00323-f001:**
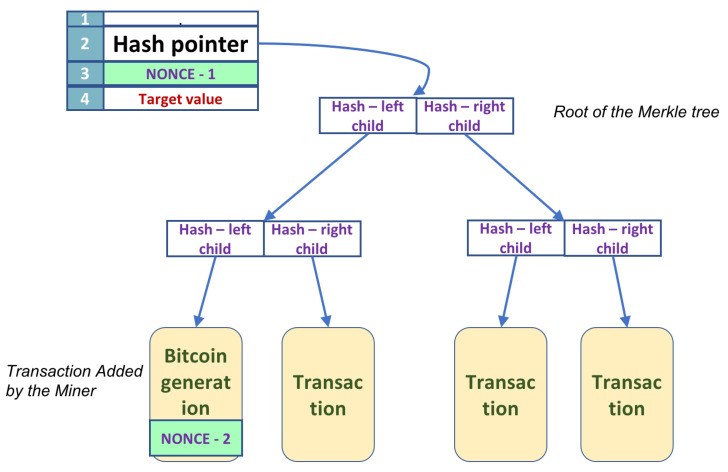
A block has a header and a Merkle tree.

**Figure 2 entropy-24-00323-f002:**
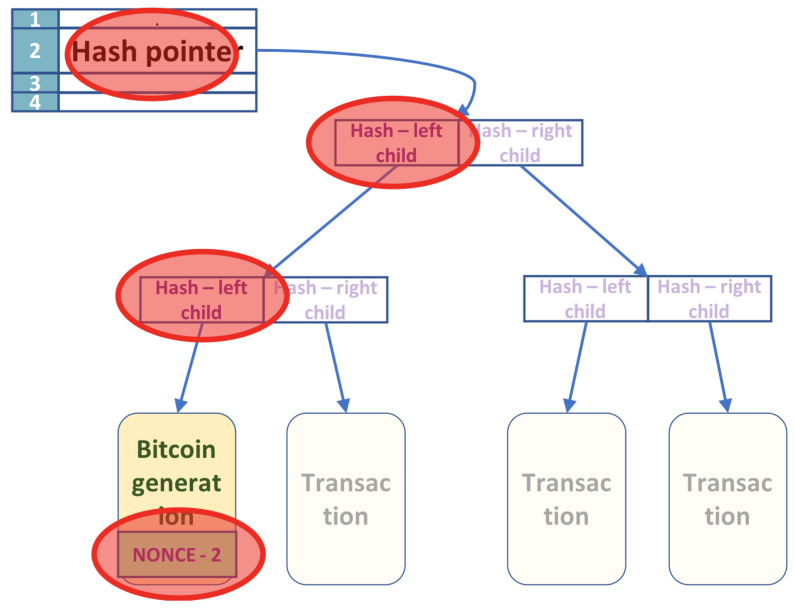
The nodes along the left leg of the tree are affected by the change of the extra nonce.

**Figure 3 entropy-24-00323-f003:**
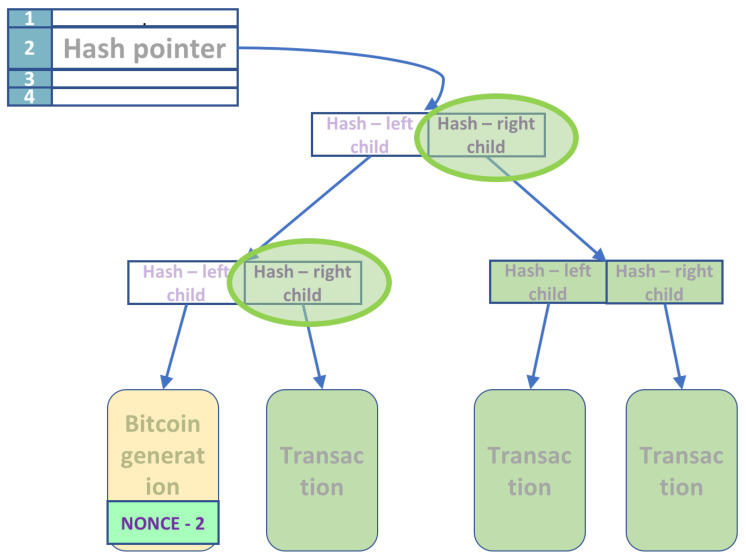
The items marked in green are all classical information. The classical circuit computes all the hash values for the right children along the Merkle tree’s left leg.

**Figure 4 entropy-24-00323-f004:**
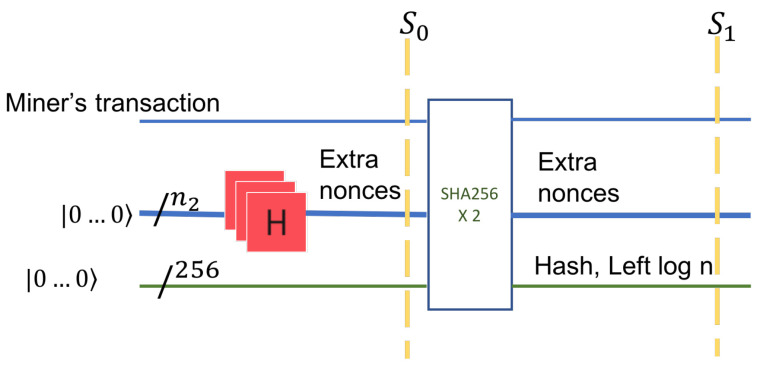
Step 2 applies the HASH function on the superposition of all possible values of the extra nonce and the miner’s classical information. Additionally, the hashing quantum circuit needs enough input qubits to hold the value of the Hash. These are 256 qubits and they are initially set to |0〉.

**Figure 5 entropy-24-00323-f005:**
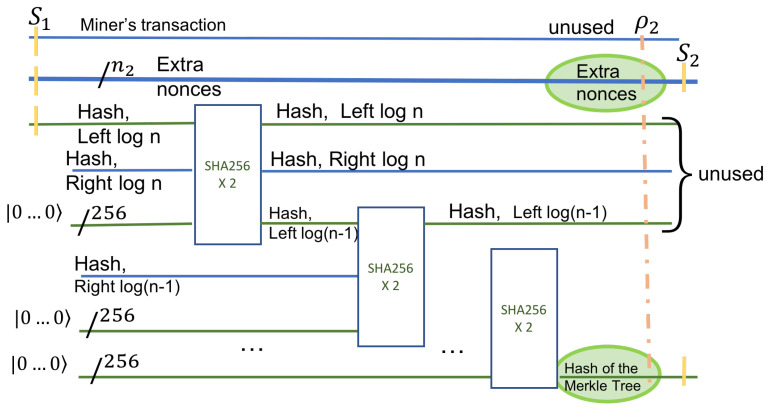
Step 3 computes the hashes along the leftmost path of the Merkle tree. All the inputs, outputs, and hashes are retrievable in the final state.

**Figure 6 entropy-24-00323-f006:**
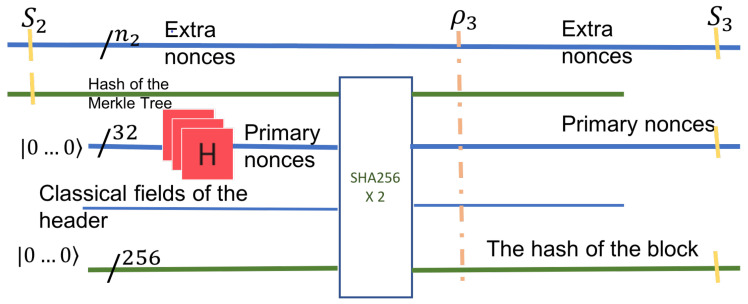
Step 4 computes the final hash value. It depends on the values of the primary nonces and the extra nonces, which are in superposition.

**Figure 7 entropy-24-00323-f007:**
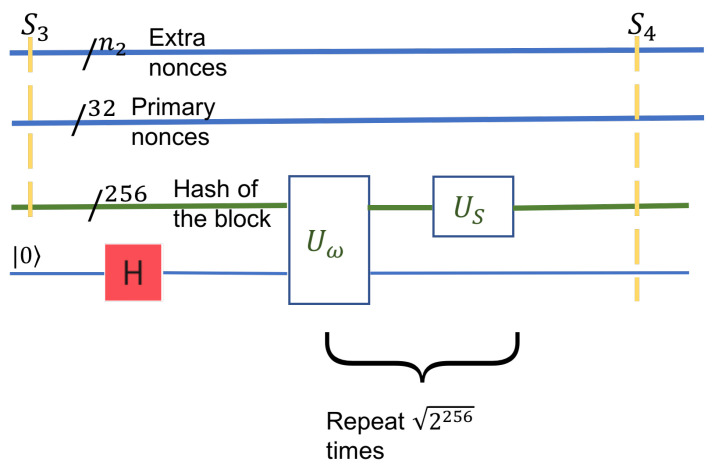
Step 5, Grover’s step.

**Figure 8 entropy-24-00323-f008:**
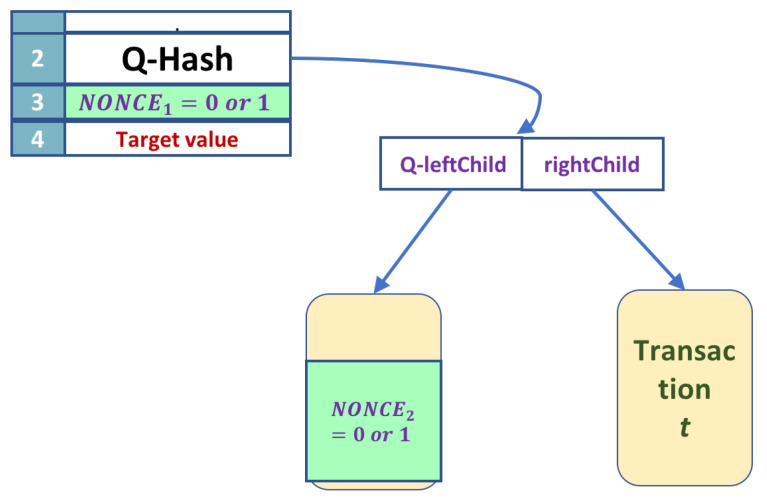
This is a small example to show the steps of the algorithm.

## Data Availability

Not applicable.

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
