# Peer review of "Quantum Bitcoin Mining"

_entropy, 2022, doi:10.3390/e24030323_

Round 1
Reviewer 1 Report
In this work, the authors proposed a classical-quantum hybrid solution to the bitcoin. The solution is divided into six stages, (1) : Compute the permanent part of the Merkle tree classically; (2) Prepare the leaf nonce in a quantum superposition and compute the first
Hash; (3) Compute all the hashes on the leftmost leg of the Merkle tree; (4) Compute the final hash, give the nonces in the block header; (5) : perform unstructured search using Grover’s algorithm; (5) measure the system to get the result. They also analyzed the cost and complexity. They expect that with 10000 qubits, the system may show quantum supremacy, which means that it will outperform the classical computer.
The problem is interesting and have very important influence in the bitcoin community. Though I had heard the threat of quantum computers to bitcoin, this is the first algorithm I see for the detailed implementation, and with specific numbers, such as the 10,000 qubits for achieving quantum supremacy. I can recommend its acceptance after revision.
The algorithm is difficult to understand. I suggest to give an example with a toy bitcoin model, which uses the smallest number of qubits as possible, as long as it can explain the basic idea of the algorithm presented in this paper, in the six steps. For instance, give apecific values to the hash pointer, nonce 1 and so on.
In recent years, the study of use of quantum computing in finance is increasing, for instance for pricing debt obligation in ref. [r1]. Layered structure quantum computing is also proposed for image recognition, which is similar to finding index in bitcoin, as in ref. [r2].
In the present study, the analysis is based on ideal qubit and deep depth of circuit. However, at present, we have only noisy qubits, like in ref.[r3], it will influence all the quantum stages. Therefore, the analysis on quantum supremacy should consider these influences, the increase in the number of qubits will be inevitable. In addition to the noises, there is also a small probability of failure in the Grover algorithm itself, which can be solved by replacing the phase inversions with some special phase rotations given in Ref. [r4]. The detailed performance of the quantum search algorithm has been extensively studied in ref. [r5].
[r1] Tang H, Pal A, Wang T Y, et al. Quantum computation for pricing the collateralized debt obligations[J]. Quantum Engineering, 2021, 3(4): e84.
[r2] Wei S J, Chen Y H, Zhou Z R, et al. A quantum convolutional neural network on NISQ devices[J]. AAPPS Bulletin, 2022, 32(1): Art. No. 2.
[r3] Zhou P, Lv L, Ming He L. Effect of noise on remote preparation of an arbitrary single‐qubit state[J]. Quantum Engineering, 2021: e64.
[r4] Long G L. Grover algorithm with zero theoretical failure rate[J]. Physical Review A, 2001, 64(2): 022307.
[r5] Toyama F M, Van Dijk W, Nogami Y. Quantum search with certainty based on modified Grover algorithms: optimum choice of parameters[J]. Quantum information processing, 2013, 12(5): 1897-1914.
Author Response
Dear Reviewer,
Thank you for your time and expertise in doing this review for our paper. We appreciate the depth of your comments concerning the content of our paper and also the request for a small explanatory example. Here are detailed answers based on your comments.
The algorithm is difficult to understand. I suggest to give an example with a toy bitcoin model, which uses the smallest number of qubits as possible, as long as it can explain the basic idea of the algorithm presented in this paper, in the six steps. For instance, give apecific values to the hash pointer, nonce 1 and so on.
We added an example on a small Merkle tree with the root and two leaves. We also considered, as you suggested, each nonce to be of size 1. This description is now the subsection 4.1 starting on page 11.
In recent years, the study of use of quantum computing in finance is increasing, for instance for pricing debt obligation in ref. [r1]. Layered structure quantum computing is also proposed for image recognition, which is similar to finding index in bitcoin, as in ref. [r2].
We added the info about these references to the Introduction.
In the present study, the analysis is based on ideal qubit and deep depth of circuit. However, at present, we have only noisy qubits, like in ref.[r3], it will influence all the quantum stages. Therefore, the analysis on quantum supremacy should consider these influences, the increase in the number of qubits will be inevitable. In addition to the noises, there is also a small probability of failure in the Grover algorithm itself, which can be solved by replacing the phase inversions with some special phase rotations given in Ref. [r4]. The detailed performance of the quantum search algorithm has been extensively studied in ref. [r5].
We believe that this is an important issue. We changed the statement in the abstract to mention explicitly that the requirement of 10^4 qubits applied on ideal qubits. We also added a paragraph at the end of the subsection 5.1 Quantum Supremacy. The paragraph is on page 16. One difficulty in actually evaluating numerically the additional number of qubits comes from the fact that it depends on the implementation of the qubit. For example, in [r3], the probability of noise effects are denoted with p and then the analysis depends on this value for all types of noise. p does not have a fixed value and depends on the implementation. We mentioned this problem in the paragraph that we added to our paper.
[r1] Tang H, Pal A, Wang T Y, et al. Quantum computation for pricing the collateralized debt obligations[J]. Quantum Engineering, 2021, 3(4): e84.
[r2] Wei S J, Chen Y H, Zhou Z R, et al. A quantum convolutional neural network on NISQ devices[J]. AAPPS Bulletin, 2022, 32(1): Art. No. 2.
[r3] Zhou P, Lv L, Ming He L. Effect of noise on remote preparation of an arbitrary single‐qubit state[J]. Quantum Engineering, 2021: e64.
[r4] Long G L. Grover algorithm with zero theoretical failure rate[J]. Physical Review A, 2001, 64(2): 022307.
[r5] Toyama F M, Van Dijk W, Nogami Y. Quantum search with certainty based on modified Grover algorithms: optimum choice of parameters[J]. Quantum information processing, 2013, 12(5): 1897-1914.
Thank you again and we hope this answers your requests for our paper.

Reviewer 2 Report
In this manuscript, the authors proposed a quantum Bitcoin mining based on Grover’s algorithm. Although they require approximately ten thousands qubits, the analysis has shown that the proposed method can quadratically reduce the execution time of finding the nonce. Note that the "stable state" in page 3 line 8 should be "stale state" and the step-by-step procedure in page 11 should remove the number labels. Besides the above minor editing issues, I would like to recommend it for publication in the Entropy.
Author Response
Dear Reviewer,
Thank you for your time and expertise in doing this review for our paper. Here are detailed answers based on your comments.
In this manuscript, the authors proposed a quantum Bitcoin mining based on Grover’s algorithm. Although they require approximately ten thousands qubits, the analysis has shown that the proposed method can quadratically reduce the execution time of finding the nonce. Note that the "stable state" in page 3 line 8 should be "stale state" and the step-by-step procedure in page 11 should remove the number labels. Besides the above minor editing issues, I would like to recommend it for publication in the Entropy.
The typo on page 3 has been corrected. The number labels are made consistent with the figures.

Round 2
Reviewer 1 Report
The revised version has improved significantly. However, there are still errors in the manuscript, especially the references list. Some references missed the journal title, or the volume or page number. For instance
Tang, Hao, et al. "Quantum computation for pricing the collateralized debt obligations." Quantum Engineering 3(4) (2021): e84.
Zhou, Ping, Li Lv, and Liang Ming He. "Effect of noise on remote preparation of an arbitrary single‐qubit state." Quantum Engineering 3(2) (2021): e64.
Wei, ShiJie, et al. "A quantum convolutional neural network on NISQ devices." AAPPS Bulletin 32(1) (2022): Art. number 2.
I recommend acceptance provided the typos are corrected.